# Toward Trustworthy: A Method for Detecting Fine-Tuning Origins in LLMs

## Abstract

As large language models (LLMs) continue to advance, their deployment often involves fine-tuning to enhance performance on specific downstream tasks. However, this customization is sometimes accompanied by misleading claims about the origins, raising significant concerns about transparency and trust within the open-source community. Existing model verification techniques typically assess functional, representational, and weight similarities. However, these approaches often struggle against obfuscation techniques, such as permutations and scaling transformations, that obscure a model's lineage. To address this limitation, we propose a novel detection method that rigorously determines whether a model has been fine-tuned from a specified base model. This method includes the ability to extract the LoRA rank utilized during the fine-tuning process, providing a more robust verification framework. This framework is the first to provide a formalized approach specifically aimed at pinpointing the sources of model fine-tuning. We empirically validated our method on twenty-nine diverse open-source models under conditions that simulate real-world obfuscation scenarios. We empirically analyze the effectiveness of our framework and finally, discuss its limitations. The results demonstrate the effectiveness of our approach and indicate its potential to establish new benchmarks for model verification.

## 1 Introduction

Recently, as large language models (LLMs) continue to advance, increasingly powerful models are rapidly emerging, demonstrating exceptional performance across a wide range of tasks. Users frequently fine-tune these models to enhance their performance for specific applications. However, certain model providers have engaged in deceptive practices, exaggerating their technological capabilities for unjust gain. For example, the Reflection-70B, marketed by HyperWrite as the worlds leading open-source model, was in fact fine-tuned on Llama3-70B-instruct, not on Llama3.1-70B as originally claimed, as illustrated in Figure 1. Such false claims have raised significant concerns regarding the potential misuse of models and the spread of misleading information (Pan et al., 2023).

Current detection methods mainly evaluate functional, representation and weight similarity, as well as training data properties and program similarity (Klabunde et al., 2023b). However, the criteria used in these methods lack the necessary rigor and formalization, leading to ambiguity and inconsistency in determining whether a model is a fine-tuned derivative of a specific base model. Among these techniques, weight similarity is considered the most effective indicator to verify the relationship between models. However, when the model undergoes obfuscation techniques such as permutation or scaling transformations (Zhou et al., 2023; Lee et al., 2018), its reliability is compromised. This shortcoming highlights the urgent need for more robust and systematic detection frameworks that can reliably identify fine-tuned models even when intentional obfuscation is involved.

To address this challenge, our study introduces a novel detection method that can rigorously determine whether a model has been fine-tuned from a specified base model. Our approach is the first formal framework designed to address the complexity of model fine-tuning for detection, marking a significant advance over existing techniques. Crucially, the method remains valid regardless of the permutations used, enabling accurate determination of the basis model for any derivative. Through this research, we aim to establish new standards for model verification in the open-source community and improve the transparency and trustworthiness of the sources of AI models.

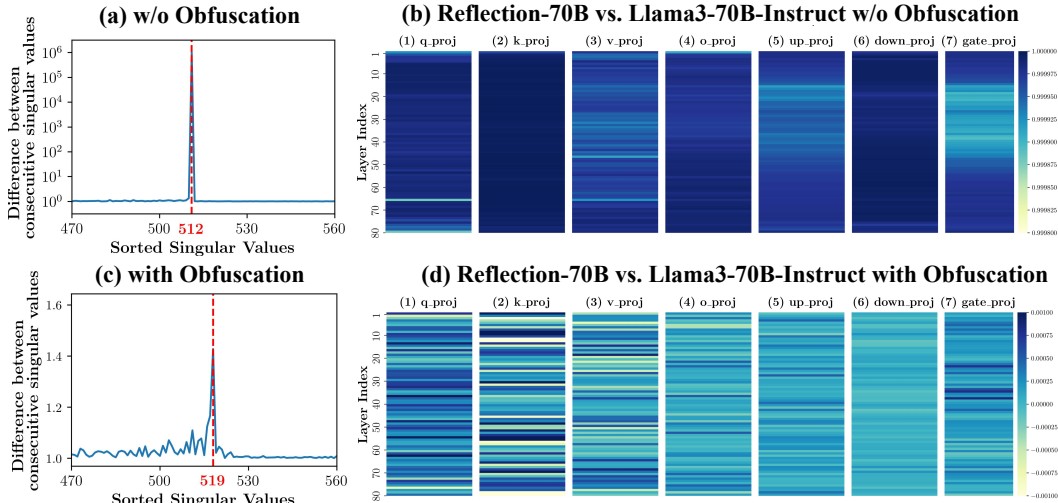

Figure 1: The detection of Reflection-70B with (w/o) obfuscation. (a) and (c) show the distinct peak in singular value differences near the rank, both without and with obfuscation. (b) and (d) depict parameter similarities across various modules when compared to Llama3-70B-Instruct, without and with obfuscation, respectively.

To empirically validate the efficacy of our detection method, we conducted tests on a diverse set of twenty-nine open-source models. Recognizing the presence of rotational transformations, we treated the model parameters as inherently unknowable, approaching each model as a gray box where only the inputs and outputs of each layer are accessible. This perspective ensures that our testing conditions reflect practical limitations typically encountered in real-world scenarios. Under these constraints, our results demonstrate that our algorithm robustly identifies fine-tuning across all tested models, confirming its broad applicability and effectiveness.

## 2 RELATED WORKS

**Parameter-Efficient Fine-Tuning.** PEFT has emerged as a crucial strategy for optimizing LLMs for specific tasks while reducing resource consumption. Techniques such as Low-Rank Adaptation (LoRA) (Hu et al., 2021; Dettmers et al., 2024), Adapter Layers (Karimi Mahabadi et al., 2021), and Prompt Tuning (Jia et al., 2022) achieve performance improvements by modifying only a small subset of parameters, thus capturing task-specific information while retaining the original model's foundational knowledge. However, the increasing reliance on these methods raises concerns about transparency and traceability, highlighting the need for robust verification techniques to ensure the integrity and reliability of fine-tuned models.

**Obfuscation Techniques.** To bolster model privacy and hinder unauthorized access, techniques such as permutation, scaling, and noise addition are employed Elhage et al. (2021). These methods obscure direct parameter comparisons, complicating the identification of derived models. For example, permutation rearranges parameters, scaling alters their magnitudes, and noise addition introduces random variations, all of which mask the model's characteristics. These obfuscation strategies protect intellectual property and sensitive data from unauthorized access and reverse engineering (Yousefi et al., 2023), while also preventing misuse.

**Detection Methods.** Recent researches for identifying model modifications focus on various similarities, including functional, representational, weight, training data, and procedural aspects (Klabunde et al., 2023a). Functional and representational similarities compare model outputs and internal activations, respectively, but often struggle against fine-tuning variations and obfuscation techniques like permutations and noise addition (Ethayarajh, 2019; Wu et al., 2020; Kornblith et al., 2019). Weight similarity can effectively detect model lineage but is compromised by permutation-based obfuscation (Wang et al., 2022; Elhage et al., 2021). Techniques examining training data and procedural similarities, such as influence functions, can illuminate fine-tuning practices but often require extensive datasets (Grosse et al., 2023; Shah et al., 2023). Additionally, procedural similarity offers

insights into training methods but is limited by the proprietary nature of training pipelines (Biderman et al., 2023; Zhao et al., 2023). Overall, recent approaches highlight the challenges in detecting model modifications amid sophisticated obfuscation tactics.

# 3 PRELIMINARIES

## 3.1 DECODER-ONLY TRANSFORMER ARCHITECTURE

A single decoder-layer is comprised of multi-head attention followed by a feed-forward network, shown as Figure 2. It takes as input a sequence $X_i$ and the output is likewise a sequence of vectors $Z_i$. The Self-Attention, referred to as $h_i$, and then transformed by the feed-forward network $f_i$, resulting in the final output as:

$$Z_i = f_i \circ h_i(X_i)$$

In Self-Attention, each attention head results in $W_Q$, $W_K$, $W_V$ and $W_O$ as the query, key, value, and output matrices respectively, which apply a linear transformation to each $x \in X$. $R_\theta$ represents RoPE, and the output equation becomes $Y_i$ as follows:

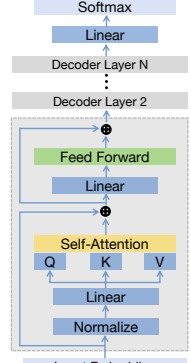

$$Y_i = \text{softmax}\left(\frac{h_{\text{norm}}(X_i)W_Q W_K^\top h_{\text{norm}}^\top(X_i)}{\sqrt{d_K}}\right) h_{\text{norm}}(X_i)W_V W_O + X_i$$

Considering feed forward network and the residual connection, the following set of equations characterizes the function of a single decoder-layer. $\sigma$ denotes an activation function (e.g., GeLU or SiLU).

Figure 2: Decoder-only Architecture

$$Z_i = [\sigma(h_{\text{norm}}(Y_i)W_G) \odot (h_{\text{norm}}(Y_i)W_{\text{up}})]W_{\text{down}} + Y_i,$$

## 3.2 OBFUSCATION

Obfuscation in neural networks refers to the deliberate manipulation of parameters or structures to obscure their original form while retaining functional output. In the context of models, obfuscation techniques such as rearranging parameter matrices in attention and MLP modules are employed to complicate unauthorized access, prevent direct comparisons, and protect model privacy. Despite the internal changes, these techniques ensure that the models functional behavior remains intact, preserving performance while safeguarding intellectual property.

Significant studies, such as those by Maron et al. (2020) and Zaheer et al. (2017), have explored obfuscation's effects in maintaining consistent outputs across different configurations. These works highlight how obfuscation stabilizes model performance and hinders reverse engineering efforts.

Mathematically, given a set $S = \{s_1, s_2, \ldots, s_n\}$, obfuscation is defined through transformations that render the underlying structure opaque. When applied to a weight matrix $W \in \mathbb{R}^{m \times n}$, transformation matrices $P \in \{0,1\}^{n \times n}$ reorder elements, turning $W$ into $WP$. These transformations complicate direct analysis without affecting the models output.

In Transformer architectures, the MLP and attention layers, denoted by $F$ and $H$, undergo obfuscation through transformations $\Pi_1$ and $\Pi_2$, defined as follows:

$$F_{\text{obf}} \circ H_{\text{obf}}(X) = F \circ H(X),$$

where $F_{\text{obf}} = \Pi_1(F)$ and $H_{\text{obf}} = \Pi_2(H)$. This approach ensures that internal obfuscation does not affect the overall output, maintaining the model's integrity and safeguarding its internal structure.

## 3.3 PROBLEM FORMULATION

This research aims to determine whether the candidate model $M_c$ has undergone fine-tuning in its self-attention modules, excluding MLP modules, from the base model $M_b$ using Low-Rank Adaptation (LoRA), followed by layer-level obfuscation. We consider both models as white boxes, but the obfuscations in $M_c$ complicate comparisons with $M_b$ due to potential parameter transformations that may obscure the structural relationships between their parameter matrices.

Let $M_c^*$ represent the ideally fine-tuned model derived from $M_b$ using Low-Rank Adaptation (LoRA) without any obfuscation. The candidate model $M_c$ is then generated from $M_c^*$ by implementing obfuscations to its layers. The challenge posed by this scenario is encapsulated by the discrepancy in ranks of the parameter differences, expressed as:

$$\text{rank}(W_c^* - W_b) = s \text{ but rank}(W_c - W_b) \gg s,$$

Here, $W$ represents the matrices of fine-tuned module parameters, $W_c^*$ denotes the ideally fine-tuned matrices without obfuscations, and $W_b$ represents the parameter matrices of the base model.

The primary challenge this research addresses is the determination of the original, unpermuted parameter matrix $W_c^*$ given the observed permuted matrix $W_c$. Our primary objective is to develop methodologies by which the structure of $W_c^*$ can be accurately inferred from $W_c$ without prior knowledge of the specific obfuscations applied.

## 4 METHODOLOGY

### 4.1 EXTRACTION OF LoRA RANK INFORMATION

In this section, we examine the extraction of low-rank information from the intermediate states, specifically the value and output projection matrices $\mathbf{W_V}$ and $\mathbf{W_O}$ in Transformer models. The intermediate state between the self-attention mechanism and the MLP layer is expressed as follows:

$$Y_i = \text{softmax}\left(\frac{h_{\text{norm}}(X_i)R_\theta W_Q W_K^\top R_\theta^\top h_{\text{norm}}^\top(X_i)}{\sqrt{d_K}}\right) h_{\text{norm}}(X_i)W_V W_O + X_i$$

where $h_{\text{norm}}(X_i)$ denotes the normalized input, $W_Q, W_K, W_V$ and $W_O$ are the query, key, value, and output projection matrices, respectively, and $R_\theta$ is a Rotational Position Encoding Matrix that incorporates positional information into the token embeddings. According to Therome 4, these parameter matrices are uniquely determined by their corresponding inputs-output pairs.

To facilitate the analysis and simplify the interpretation of the intermediate state, we focus on cases where the embedded tokens reduce to a **one-dimensional tensor**. Specifically, let the input tensor be $x \in \mathbb{R}^{1 \times d}$. Under this condition, the intermediate state simplifies as:

$$y = h_{\text{norm}}(x)W_V W_O + x$$

*Proof.* Let $x \in \mathbb{R}^{1 \times d}$. Consequently, the normalized representation $h_{\text{norm}}(x)$ remains within $\mathbb{R}^{1 \times d}$, thus preserving its one-dimensional nature. This dimensionality reduction simplifies the expression within the softmax argument. Specifically, the term:

$$\frac{h_{\text{norm}}(x)R_\theta W_Q W_K^\top R_\theta^\top h_{\text{norm}}^\top(x)}{\sqrt{d_K}}$$

collapses to a scalara constantdue to the operations involving one-dimensional vectors and matrices. When the softmax function is applied to this scalar, it simplifies to 1, given that the softmax of a scalar input reduces to a normalized value of 1:

$$\text{softmax}\left(\frac{h_{\text{norm}}(x)R_\theta W_Q W_K^\top R_\theta^\top h_{\text{norm}}^\top(x)}{\sqrt{d_K}}\right) = 1$$

Thus, the intermediate state can be represented without the softmax operation, yielding:

$$y = h_{\text{norm}}(x)W_V W_O + x$$

Let $y$ and $\tilde{y}^*$ denote the intermediate states of $M_b$ and $M_c^*$ for the same input tensor $x$. This relationship can be expressed as:

$$y - \tilde{y}^* = h_{\text{norm}}(x)\left(W_V W_O - \tilde{W}_V^* \tilde{W}_O^*\right).$$

We can simplify this to:

$$y - \tilde{y}^* = h_{\text{norm}}(x)W_{\text{low}},$$

where $W_{\text{low}} = W_V W_O - \tilde{W}_V^* \tilde{W}_O^*$ represents the difference reflecting the low-rank component. Assuming the input space $x$ spans a set of linearly independent vectors that form a full-rank matrix $X$, and based on Lemma 1, we have $\text{rank}(h_{\text{norm}}(X)) = \text{rank}(X)$. Thus,

$$Y = h_{\text{norm}}(X)W_{\text{low}}.$$

We use Singular Value Decomposition (SVD) to extract and characterize the low-rank information from the matrix $Y$, highlighting the differences between the base and fine-tuned models.

For empirical validation, we created a dataset using the Natural Language Toolkit (NLTK). A vocabulary of words was processed through the model's tokenizer and embedding layers, yielding 5,579 one-dimensional tensors. This dataset serves as the foundation for our subsequent analysis.

## 4.2 Equivalent Intermediate Reconstruction

In this section, we explore the reconstruction of intermediate states from the **output** and the **MLP** module of **base model** and address how to resolve obfuscations involved in these processes. The relationship between single decoder-layer of $M_c$ and $M_c^*$ can be formalized as:

$$f_c \circ h_c(x) = f_c^* \circ h_c^*(x) \text{ and } f_c = \Pi_2(f_c^*), h_c = \Pi_1(h_c^*),$$

where $\Pi_1$ and $\Pi_2$ are obfuscation operations applied to the MLP and attention parameters, respectively. Given that $f_c^* = f_b$, the equation simplifies to:

$$z_c = f_c \circ h_c(x) = f_b \circ h_c^*(x),$$

which implies that:

$$h_c^*(x) = f_b^{-1}(z_c).$$

Consider the equation for $z$ as follows:

$$z = [\sigma(h_{\text{norm}}(y)W_G) \odot (h_{\text{norm}}(y)W_{\text{up}})]W_{\text{down}} + y.$$

This equation describes a non-linear transformation involving both element-wise operations and matrix multiplications, rendering the inverse mapping from the output back to the input analytically intractable. Given the nonlinearity and complexity of this transformation, directly inferring the intermediate state $y$ from the observed output $z$ poses significant challenges. To tackle this, we adopt an iterative optimization strategy using gradient descent to approximate the original intermediate state $y$ that likely produced the observed output. The goal is to minimize the discrepancy between the MLP output and the actual observed output by adjusting $y$. The iterative update formula is expressed as:

$$y_{m+1} = y_m - \alpha \nabla \|f(y_m) - z_c\|^2$$

where $z_c$ denotes the layer output of $M_c$, $y_m$ denotes the estimated intermediate state at iteration $m$, $\alpha$ is the learning rate, and $\nabla \|f(y_m) - z_c\|^2$ represents the gradient of the loss function with respect to $y_m$. This loss function quantifies the squared error between the MLP output $f(y_m)$ and the target output $z_c$. By iteratively updating $y_m$, the gradient descent algorithm aims to converge on an intermediate state $y^*$ that, when processed through the MLP, closely replicates the observed output $z_c$. This reconstruction approach facilitates the approximation of hidden intermediate states from the MLP outputs, providing a mechanism to indirectly assess and compare the internal representations across different models, such as the base model $M_b$ and the candidate model $M_c$.

## 4.3 Rank Extraction

While the reconstruction process can confirm the authenticity of a base model, additional analysis is required to ascertain the peft-tuning rank during training. We approximate this rank using the reconstructed intermediates, as outlined in Algorithm 1. Here, $h$ represents the dataset size, and $n$ is chosen marginally larger than the typical maximum rank used in LoRA fine-tuning.

**Rationale** As detailed in Appendix A, the output function of a single decoder layer is bijective with probability 1. However, certain outputs that are nearly identical may correspond to intermediates that are not sufficiently similar. This poses a significant challenge in identifying which intermediates are adequately close to the true intermediate. To address this, we implement a random sampling algorithm based on the hypothesis that if outputs are nearly identical, their corresponding intermediates

---

**Algorithm 1** Random Rank Extraction

---

**Require:** Sufficiently close intermediate in the reconstructed intermediate $Y$

1: Initialize $n$ to a value less than $h$
2: Set $rank_{min}$ to the dimension of the hidden state
3: Initialize the cycle times $t$
4: **for** $i = 1$ to $t$ **do**
5:     $List[num_1, num_2, \cdots, num_n] \leftarrow$ RandChose$(n, h)$     $\triangleright$ Choose $n$ random indices from $h$
6:     $Y_i \leftarrow$ Compose$(Y, Y_b, List)$              $\triangleright$ Compose the matrix from the index list
7:     $\lambda_1 \geq \lambda_2 \geq \cdots \geq \lambda_n \leftarrow$ SingularValues$(Y_i)$
8:     $rank \leftarrow \arg\max_i (\log \|\lambda_i\| - \log \|\lambda_{i+1}\|)$
9:     $rank_{min} \leftarrow \min(rank_{min}, rank)$           $\triangleright$ Update with the minimum rank
10: **end for**
11: **return** $rank_{min}$

---

are likely similar. Assuming sufficient iterations, this method is expected to reliably approximate the true rank. The probability $P$ of achieving the true rank can be expressed as follows:

$$P = \lim_{n \to \infty} 1 - (1 - p_s)^n = 1,$$

Here, $n$ represents the number of cycles, and $p_s$ denotes the probability that all selected intermediates are adequately close to the true intermediate. Details are provided in Appendix B.

## 5 EXPERIMENTS

### 5.1 EXPERIMENTAL SETUP

**Models and Datasets.** We consider twenty-nine open-source, LoRA-fine-tuned models with various architectures as target models. Specifically, we select the series of **LLaMA2** (7B, 13B, 70B) Touvron et al. (2023), **LLaMA3** (8B, 70B), and **Mistral** (7B,13B) Jiang et al. (2023), as base models. We constructed a 5k dataset from the Natural Language Toolkit (NLTK) Loper & Bird (2002), ensuring each input was encoded as a single token to maintain a consistent attention score during the self-attention module.

**Rank Extraction Method.** We extract the LoRA rank by analyzing the differences in intermediate representations between the target model and its base model. Then, we compute the dimensionality of this subspace with SVD(Singular Value Decomposition). The rank is determined by a sharp drop (Figure 3) in singular values, seen as a peak (Figure 4) in the differences between consecutive values.

**Implementation Details.** To reconstruct the inputs for the MLP, we utilized gradient descent with the Adam optimizer Kingma & Ba (2014), initiating the process with a learning rate of 1.5e-3. We incorporated a **StepLR** scheduler with a step size of 1 to periodically adjust the learning rate by a factor of 0.9999, thus finely tuning the learning rate reduction to balance rapid convergence with meticulous adjustment of the model parameters. Furthermore, to address potential inaccuracies in reverse-engineered MLP inputs, we conducted 50 iterations of stochastic sampling, each consisting of 520 output evaluations. The minimum rank determined from these iterations forms our final estimation, enhancing the overall precision by reducing the influence of outlier data.

### 5.2 MAIN RESULTS

In this subsection, We apply the rank extraction algorithm to the previously considered models, demonstrating its effectiveness in finding the rank of LoRA tuning.

**Effectiveness on Models.** We implemented rank extraction across all layers for each model. To determine the most representative rank, we systematically selected the smallest value from the extracted ranks for each layer. This methodology ensures that our final results reflect the minimal dimension necessary to capture the underlying transformations within the model. The comprehensive results of this analysis are summarized in Table 1. To visually represent the critical points in our analysis, we include two figures, figure 3 illustrates the sharp drop in singular values observed across different models and Figure 4 displays the peaks in the ratios of consecutive singular values.

Table 1: Extraction results across a range of LoRA-fine-tuned models, applying LoRA to different types of attention weights in models. (Given Rank | Extracted Rank)

| Base | Size | Target | LoRA Config | Given rank | Extracted Rank |
|---|---|---|---|---|---|
| **Llama3.1** | **8B** | Souththzz | $W_q,W_v$ | 8 | 8 |
| | | Fdelduchetto | $W_q,W_v$ | 16 | 19 |
| | | Anthonysicilia | $W_q,W_v$ | 32 | 35 |
| | | Faridlazuarda | $W_q,W_v$ | 64 | 67 |
| | | Dror44 | $W_q,W_v$ | 128 | 128 |
| | **70B** | RikiyaT | $W_q,W_v$ | 16 | 16 |
| **Llama3** | **8B** | SwastikM | $W_q,W_v$ | 8 | 8 |
| | | Islam23 | $W_q,W_k,W_v,W_o$ | 16 | 32 |
| | | Namespace-Pt | $W_q,W_k,W_v,W_o$ | 32 | 67 |
| | | Nutanix | $W_q,W_k,W_v$ | 64 | 67 |
| | | Decision-oaif | $W_q,W_v$ | 128 | 133 |
| | **70B** | ScaleGenAI | $W_q,W_v$ | 8 | 8 |
| | | Reflection-Llama | $W_q,W_v$ | 512 | 517 |
| **Llama2** | **7B** | FinGPT-7B | $W_q,W_k,W_v$ | 8 | 8 |
| | | Junhaos-nv | $W_q,W_v$ | 16 | 18 |
| | | Renyiyu | $W_q,W_v$ | 32 | 34 |
| | | Dtthanh | $W_q,W_v$ | 64 | 66 |
| | | RuterNorway-7B | $W_q,W_v$ | 128 | 128 |
| | **13B** | FinGPT-13B | $W_q,W_k,W_v$ | 8 | 8 |
| | | Lajonbot | $W_q,W_v$ | 16 | 16 |
| | | RuterNorway-13B | $W_q,W_v$ | 32 | 32 |
| | | Blackroot | $W_q,W_v$ | 64 | 64 |
| | | Zayjean | $W_q,W_v$ | 256 | 256 |
| | **70B** | Yukang | $W_q,W_k,W_v,W_o$ | 8 | 16 |
| **Mistral** | **7B** | CleverShovel | $W_q,W_v$ | 8 | 8 |
| | | BlazeLlama | $W_q,W_v$ | 16 | 16 |
| | | paragdakle | $W_q,W_k,W_v,W_o$ | 32 | 64 |
| | | Farmnetz | $W_q,W_k,W_v,W_o$ | 64 | 128 |
| | | paragdakle | $W_q,W_k,W_v,W_o$ | 128 | 256 |

**Difference on Layers.** Our findings indicate a pronounced variance in the efficacy of the intermediate state reconstruction algorithm across different layers of the model. Notably, the performance in the middle layers significantly surpasses that observed in the initial and final layers, as illustrated in Figure 5. This observation underscores the importance of layer-specific dynamics in the effectiveness of model reconstruction techniques.

## 5.3 DISCUSSION

**Accelerate.** To accelerate the progress so that the reconstruction process of the intermediate state to 700 iterations. This decision was based on utilizing the intermediate state of the base model as the initial condition, owing to its substantial similarity to the target model's intermediate. This similarity led to a small loss at the outset of training, allowing for rapid convergence shown in Figure 6. Following the initial phase of convergence using a step-based learning rate adjustment (StepLR). To adjust the learning rate more finely, after each iteration, the learning rate becomes 0.9999 times the original. This

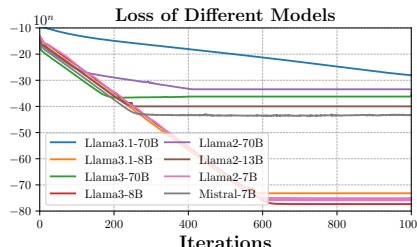

Figure 6: Loss decline curve of each base model.

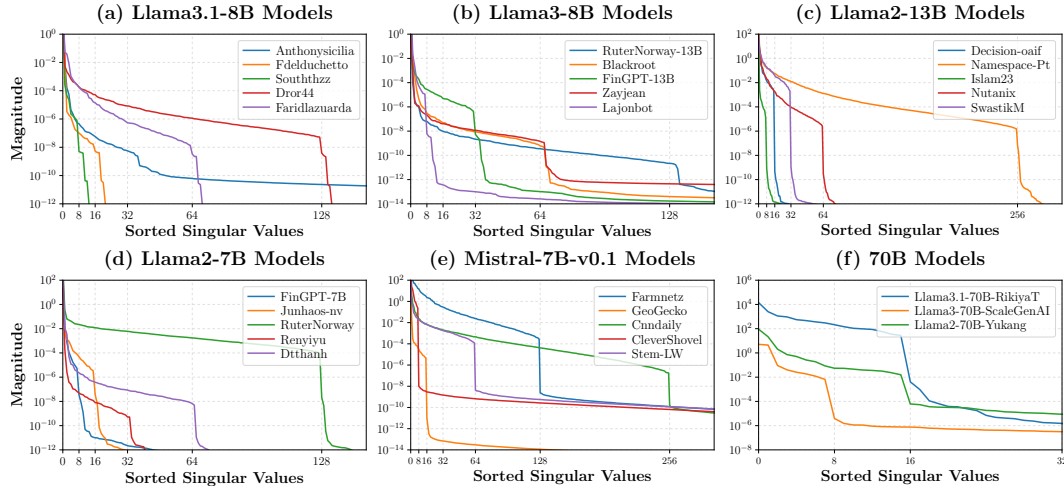

Figure 3: SVD can capture the LoRA rank of the model when the quantity of output vectors exceeds the LoRA rank. In this paper, we extract the LoRA rank of the models by accurately determining the rank through the analysis of 520 output vectors.

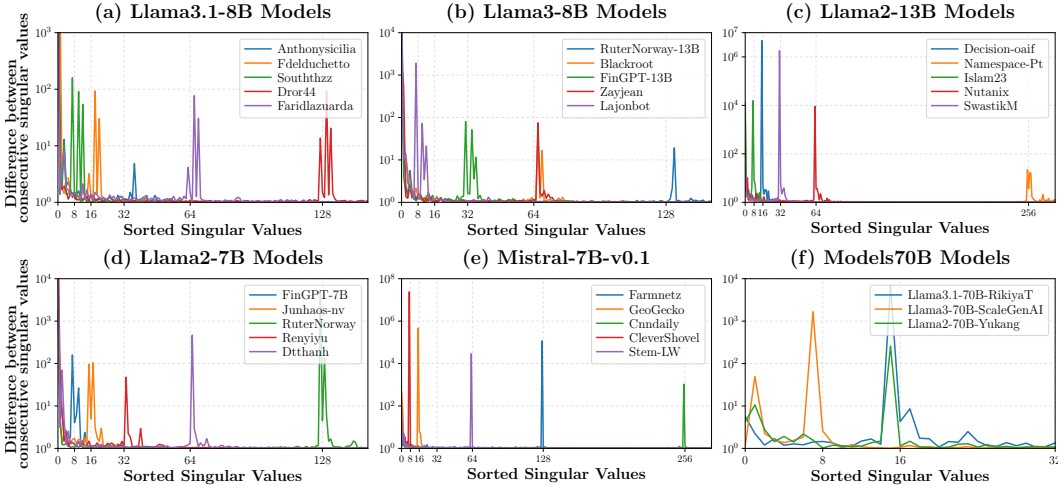

Figure 4: Our detection method determines the LoRA rank by pinpointing a sharp decline in singular values, which manifests as a peak in the disparity between consecutive singular values. In the model, this peak occurs at a position adjacent to the rank.

micro-adjustment aimed to refine the alignment between our reconstructed intermediate and the target models true intermediate, enhancing the precision of our results.

**Layer Selection.** In the previous section, we found that the effectiveness of our algorithm for computing the LoRA (Low-Rank Adaptation) rank is significantly greater in the intermediate layers than in the front and rear layers. Experimental results indicate that rank estimates from intermediate layers are closer to the true low-rank structure, highlighting their critical role in the model.

Choosing outputs from intermediate layers for approximation provides an effective means to evaluate rank calculation performance in a label-free context. By quantifying the 2-norm of the intermediate layer outputs, we can assess the effectiveness of different layers without relying on explicitly labeled data, thus identifying optimal layers for reconstruction.

Results as shown in Figure 7, demonstrate that the 2-norm of intermediate layer outputs is significantly higher than that of the front and rear layers, with rank estimates closely aligning with true values. This may be due to the inherent characteristics of intermediate layers, which better convey information when processing complex data.

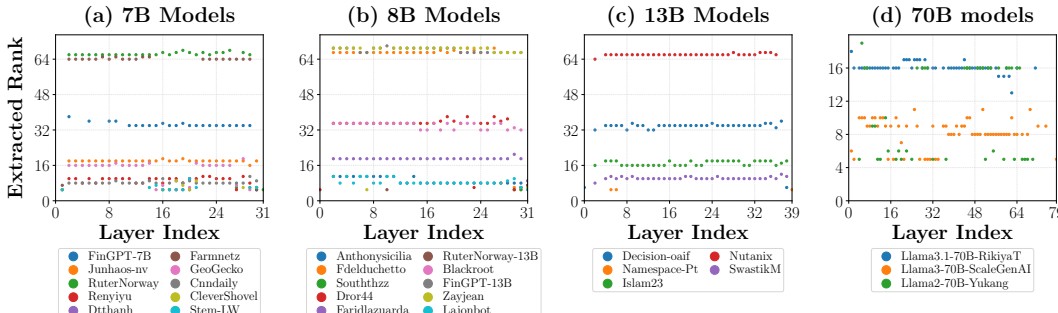

Figure 5: Rank Extraction Across Layers in Various Models. The extracted ranks for each layer within different target models are categorized by their size, 7B Models(a), 8B Models(b), 13B Models(c), and 70B Models(d), illustrate the distribution of extracted ranks across the layers of respective model architectures. Each point represents the extracted rank for a specific layer.

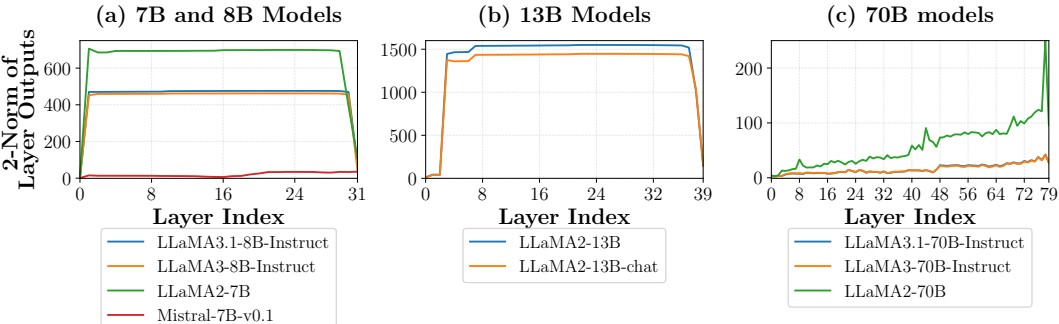

Figure 7: Norm of Layer Outputs Across Model Architectures. This figure presents the L2 norms of outputs across layers in models of varying sizes, 7B and 8B models(a), 13B models(b), and 70B models(c).

## 6 CONCLUSION

This study addresses growing concerns within the open-source community about the misrepresentation and misuse of fine-tuned models. Existing detection methods, while valuable, often struggle against sophisticated obfuscation techniques like permutations and scaling transformations, complicating the verification of modified models' lineage and authenticity. We introduce a novel detection methodology aimed at overcoming these challenges through a rigorous and systematic approach to ascertain the provenance of fine-tuned models.

Our methodology establishes a new benchmark for transparency and reliability in managing open-source models. By implementing this formalized framework, we enhance the trustworthiness of the ecosystem, ensuring that the origins and modifications of models are accurately documented, thereby promoting greater accountability in AI technology deployment.

**Limitations.** Despite its effectiveness, our method has limitations that restrict its broader applicability. It is currently designed for scenarios where MLP layers remain unmodified during fine-tuning; modifications to these layers, whether through parameter adjustments or architectural changes, reduce the effectiveness of our detection capabilities in complex model configurations. Future research will aim to extend this technique to accommodate various fine-tuning strategies, particularly those impacting MLP layers. Additionally, our method struggles with models exhibiting small output norms, which hampers the efficiency of reverse engineering intermediate states due to weakened gradient signals during reconstruction. This highlights the need for further refinement of the algorithm to ensure robust performance across varying output magnitudes.

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

## A FORMULATION DETAIL

**Lemma 1.** *For a given $X \in \mathbb{R}^{n \times d}$, consider the function $h_{norm} : \mathbb{R}^{n \times d} \to \mathbb{R}^{n \times d}$ defined as*

$$
h_{norm}(X) = \begin{pmatrix} \frac{\mathbf{x}_1}{\sqrt{\frac{1}{d}\sum_{j=1}^{d} x_{1j}^2 + \epsilon}} \\ \frac{\mathbf{x}_2}{\sqrt{\frac{1}{d}\sum_{j=1}^{d} x_{2j}^2 + \epsilon}} \\ \vdots \\ \frac{\mathbf{x}_n}{\sqrt{\frac{1}{d}\sum_{j=1}^{d} x_{nj}^2 + \epsilon}} \end{pmatrix} \odot \gamma
$$

*where $\epsilon$ is a small positive scalar and $\gamma \in \mathbb{R}^{1 \times d}$ with $\gamma \neq \mathbf{0}_d^\top$. $h_{norm}$ is bijective for non-parallel vectors.*

*Proof.* We prove it by contradiction, we need to demonstrate that if $X$ and $Y$ are two matrices in $\mathbb{R}^{n \times d}$ such that $x_i \nparallel y_i$, then $h_{\text{norm}}(X)$ and $h_{\text{norm}}(Y)$ cannot be equal.

**Injectivity:** Assume by contradiction that $h_{\text{norm}}(X) = h_{\text{norm}}(Y)$ for some $X, Y \in \mathbb{R}^{n \times d}$ such that $x_i \nparallel y_i$, by the definition of $h_{\text{norm}}$, we have:

$$
\begin{pmatrix} \frac{\mathbf{x}_1}{\sqrt{\frac{1}{d}\sum_{j=1}^{d} x_{1j}^2 + \epsilon}} \\ \frac{\mathbf{x}_2}{\sqrt{\frac{1}{d}\sum_{j=1}^{d} x_{2j}^2 + \epsilon}} \\ \vdots \\ \frac{\mathbf{x}_n}{\sqrt{\frac{1}{d}\sum_{j=1}^{d} x_{nj}^2 + \epsilon}} \end{pmatrix} \odot \gamma = \begin{pmatrix} \frac{\mathbf{y}_1}{\sqrt{\frac{1}{d}\sum_{j=1}^{d} y_{1j}^2 + \epsilon}} \\ \frac{\mathbf{y}_2}{\sqrt{\frac{1}{d}\sum_{j=1}^{d} y_{2j}^2 + \epsilon}} \\ \vdots \\ \frac{\mathbf{y}_n}{\sqrt{\frac{1}{d}\sum_{j=1}^{d} y_{nj}^2 + \epsilon}} \end{pmatrix} \odot \gamma,
$$

which implies that

$$
\left( \frac{x_{ik}}{\sqrt{\frac{1}{d}\sum_{j=1}^{d} x_{ij}^2 + \epsilon}} - \frac{y_{ik}}{\sqrt{\frac{1}{d}\sum_{j=1}^{d} y_{ij}^2 + \epsilon}} \right) \gamma_k = 0 \quad \text{for all } i, k.
$$

Further

$$
\left( \frac{x_{ik}}{\sqrt{\frac{1}{d}\sum_{j=1}^{d} x_{ij}^2 + \epsilon}} - \frac{y_{ik}}{\sqrt{\frac{1}{d}\sum_{j=1}^{d} y_{ij}^2 + \epsilon}} \right) = 0 \quad \text{for all } i, k.
$$

This implies that $x_i$ and $y_i$ must be parallel, which is a contradiction. Thus, $h_{\text{norm}}$ is injective for non-parallel vectors.

**Surjectivity:** To prove surjectivity, we must show that for any non-zero line matrix $Z \in \mathbb{R}^{n \times d}$, there exists an $X \in \mathbb{R}^{n \times d}$ such that $h_{\text{norm}}(X) = Z$.

Consider the equation:

$$Z = \begin{pmatrix} \frac{\mathbf{x}_1}{\sqrt{\frac{1}{d} \sum_{j=1}^{d} x_{1j}^2 + \epsilon}} \\ \frac{\mathbf{x}_2}{\sqrt{\frac{1}{d} \sum_{j=1}^{d} x_{2j}^2 + \epsilon}} \\ \vdots \\ \frac{\mathbf{x}_n}{\sqrt{\frac{1}{d} \sum_{j=1}^{d} x_{nj}^2 + \epsilon}} \end{pmatrix} \odot \gamma.$$

We can solve for $X$ as:

$$X = \left( Z \odot \frac{\gamma^{-1}}{\sqrt{\frac{1}{d} \sum_{i=1}^{d} X_i^2 + \epsilon}} \right).$$

Given any $Z$, we can choose $X$ such that the above equation holds. Since $\gamma$ is non-zero and $\epsilon$ is a small positive scalar, it is always sure to find such an $X$, proving that $h_{\text{norm}}$ is surjective.

So $h_{\text{norm}}$ is bijective for non-parallel vectors. □

**Lemma 2.** *For a given $x \in \mathbb{R}^{n \times d}$, function $f : \mathbb{R}^{n \times d} \to \mathbb{R}^{n \times d}$ defined as*

$$f(x) = [\sigma(h_{norm}(x)W_G) \odot (h_{norm}(x)W_{up})]W_{down} + x,$$

*where $W_G, W_{up} \in \mathbb{R}^{d \times p}$ and $W_{down} \in \mathbb{R}^{p \times d}$ are all full rank matrices and $p > d$. Additionally, $h_{norm}(X)$ is defined that*

$$h_{norm}(X) = \frac{X}{\sqrt{\frac{1}{d} \sum_{i=1}^{d} X_i^2 + \epsilon}} \odot \gamma.$$

*The function $f$ is injective for non-parallel vectors.*

*Proof.* To prove that $f(X)$ is injective for non-parallel vectors, we need to show that if $f(X) = f(Y)$, then $X = Y$ for any $X, Y \in \mathbb{R}^{n \times d}$ such that $X_i \nparallel Y_i$.

Assume that $f(X) = f(Y)$. This implies:

$$[\sigma(h_{\text{norm}}(X)W_G) \odot (h_{\text{norm}}(X)W_{up})]W_{\text{down}} + X = [\sigma(h_{\text{norm}}(Y)W_G) \odot (h_{\text{norm}}(Y)W_{up})]W_{\text{down}} + Y.$$

By the Lemma 1, we can simplify the equation and rearranging it, we have:

$$[\sigma(XW_G) \odot (XW_{up})]W_{\text{down}} - [\sigma(YW_G) \odot (YW_{up})]W_{\text{down}} = Y - X.$$

Suppose that the token space $\mathcal{C}$ is countable, which means that $f : \mathcal{C}^n \to \mathcal{C}^n$. Let

$$\mathcal{M}_{ij} = \{(W_G, W_{up}, W_{down}) | [\sigma(C_i W_G) \odot (C_i W_{up})]W_{down} = [\sigma(C_j W_G) \odot (C_j W_{up})]W_{down}\},$$

where

$$C_i \in \mathcal{C} \quad \text{and} \quad C_j \in \mathcal{C},$$

then we can get that

$$dim(\mathcal{M}_{ij}) = 3dp - nd < 3dp,$$

which means that the Lebesgue measure of $\mathcal{M}_{ij}$

$$\mu_L(\mathcal{M}_{ij}) = 0.$$

Suppose that

$$\mathcal{M} = \bigcup_{i,j} \mathcal{M}_{ij},$$

this indicates that

$$\mu_L(\mathcal{M}) = 0.$$

So we can show that

$$(W_G, W_{up}, W_{down}) \notin \mathcal{M} \quad \text{with probability 1.}$$

Hence, we can say function $f$ is injective for non-parallel vectors. □

**Lemma 3.** *Let $A$ and $B$ be distinct matrices of the same dimension, i.e., $A, B \in \mathbb{R}^{m \times n}$. Then, we have that*

$$softmax(A) = softmax(B)$$

*indicates that*

$$(A - B) = \begin{pmatrix} \alpha_1 \\ \alpha_2 \\ \vdots \\ \alpha_m \end{pmatrix} \mathbf{1}_n^\top,$$

*where $\mathbf{1}_n$ is the column vector of ones of length $n$ and $\alpha_1, \ldots, \alpha_m$ are scalars.*

*Proof.* Let matrices $A$ and $B$ be defined as follows:

$$A = \begin{pmatrix} \mathbf{a}_1 \\ \mathbf{a}_2 \\ \vdots \\ \mathbf{a}_m \end{pmatrix} = \begin{pmatrix} a_{11} & a_{12} & \cdots & a_{1n} \\ a_{21} & a_{22} & \cdots & a_{2n} \\ \vdots & \vdots & \vdots & \vdots \\ a_{m1} & a_{m2} & \cdots & a_{mn} \end{pmatrix}, \quad B = \begin{pmatrix} \mathbf{b}_1 \\ \mathbf{b}_2 \\ \vdots \\ \mathbf{b}_m \end{pmatrix} = \begin{pmatrix} b_{11} & b_{12} & \cdots & b_{1n} \\ b_{21} & b_{22} & \cdots & b_{2n} \\ \vdots & \vdots & \vdots & \vdots \\ b_{m1} & b_{m2} & \cdots & b_{mn} \end{pmatrix}.$$

The softmax function applied to each row vector $\mathbf{a}_i$ of matrix $A$ is computed as follows:

$$\text{softmax}(\mathbf{a}_i) = \left( \frac{e^{a_{i1}}}{\sum_{j=1}^n e^{a_{ij}}}, \frac{e^{a_{i2}}}{\sum_{j=1}^n e^{a_{ij}}}, \cdots, \frac{e^{a_{in}}}{\sum_{j=1}^n e^{a_{ij}}} \right).$$

For matrices $A$ and $B$, the condition

$$\text{softmax}(A) = \text{softmax} \begin{pmatrix} \mathbf{a}_1 \\ \mathbf{a}_2 \\ \vdots \\ \mathbf{a}_m \end{pmatrix} = \text{softmax}(B) = \text{softmax} \begin{pmatrix} \mathbf{b}_1 \\ \mathbf{b}_2 \\ \vdots \\ \mathbf{b}_m \end{pmatrix},$$

which means that

$$\left( \frac{e^{a_{i1}}}{\sum_{j=1}^n e^{a_{ij}}}, \frac{e^{a_{i2}}}{\sum_{j=1}^n e^{a_{ij}}}, \cdots, \frac{e^{a_{in}}}{\sum_{j=1}^n e^{a_{ij}}} \right) = \left( \frac{e^{b_{i1}}}{\sum_{j=1}^n e^{b_{ij}}}, \frac{e^{b_{i2}}}{\sum_{j=1}^n e^{b_{ij}}}, \cdots, \frac{e^{b_{in}}}{\sum_{j=1}^n e^{b_{ij}}} \right).$$

This indicates that

$$a_{ij} - a_{ik} = b_{ij} - b_{ik},$$

further

$$a_{ij} - b_{ij} = a_{ik} - b_{ik}$$

Hence, we have concluded that:

$$(A - B) = \begin{pmatrix} \alpha_1 \\ \alpha_2 \\ \vdots \\ \alpha_m \end{pmatrix} \mathbf{1}_n^\top.$$

$\square$

**Theorem 4.** *Given identical inputs and outputs, the parameter matrix of a single decoder-layer is uniquely determined.*

*Proof.* Recall that the output of a single decoder-layer is the concatenation of a residual MLP $f$ and a residual self-attention $h_A$, where

$$f(X) = [\sigma(h_{\text{norm}}(X)W_G) \odot (h_{\text{norm}}(X)W_{\text{up}})]W_{\text{down}} + X, \tag{1}$$

$$\sigma(X) = \begin{pmatrix} \frac{x_{1,1}}{1+e^{-x_{1,1}}} & \frac{x_{1,2}}{1+e^{-x_{1,2}}} & \cdots & \frac{x_{1,d}}{1+e^{-x_{1,d}}} \\ \frac{x_{2,1}}{1+e^{-x_{2,1}}} & \frac{x_{2,2}}{1+e^{-x_{2,2}}} & \cdots & \frac{x_{2,d}}{1+e^{-x_{2,d}}} \\ \vdots & \vdots & \vdots & \vdots \\ \frac{x_{n,1}}{1+e^{-x_{n,1}}} & \frac{x_{n,2}}{1+e^{-x_{n,2}}} & \cdots & \frac{x_{n,d}}{1+e^{-x_{n,d}}} \end{pmatrix} \tag{2}$$

$$h_{\text{norm}}(X) = \begin{pmatrix} \frac{\mathbf{x}_1}{\sqrt{\frac{1}{d}\sum_{j=1}^d x_{1j}^2 + \epsilon}} \\ \frac{\mathbf{x}_2}{\sqrt{\frac{1}{d}\sum_{j=1}^d x_{2j}^2 + \epsilon}} \\ \vdots \\ \frac{\mathbf{x}_n}{\sqrt{\frac{1}{d}\sum_{j=1}^d x_{nj}^2 + \epsilon}} \end{pmatrix} \odot \gamma \tag{3}$$

and

$$h_A(X) = \text{softmax}\left( \frac{h_{\text{norm}}(X) R_\theta \tilde{W}_Q \tilde{W}_K^\top R_\theta^\top h_{\text{norm}}^\top(X)}{\sqrt{d_K}} \right) h_{\text{norm}}(X) \tilde{W}_V \tilde{W}_O + X, \tag{4}$$

where $X \in \mathbb{R}^{n \times d}$; $W_G$ and $W_{up} \in \mathbb{R}^{d \times g}$; $W_{down} \in \mathbb{R}^{g \times d}$; $\gamma \in \mathbb{R}^{1 \times d}$; $W_Q, W_K, W_V$ and $W_O \in \mathbb{R}^{d \times d}$,

$$R_\theta = \begin{pmatrix} \cos(m\theta_1) & -\sin(m\theta_1) & 0 & \cdots & 0 & 0 & \\ \sin(m\theta_1) & \cos(m\theta_1) & 0 & \cdots & 0 & 0 & \\ 0 & 0 & \cos(m\theta_2) & -\sin(m\theta_2) & \cdots & 0 & 0 \\ 0 & 0 & \sin(m\theta_2) & \cos(m\theta_2) & \cdots & 0 & 0 \\ \vdots & \vdots & \vdots & \vdots & \ddots & \vdots & \vdots \\ 0 & 0 & 0 & 0 & \cdots & \cos(m\theta_{d/2}) & -\sin(m\theta_{d/2}) \\ 0 & 0 & 0 & 0 & \cdots & \sin(m\theta_{d/2}) & \cos(m\theta_{d/2}) \end{pmatrix}.$$

By Lemma 2, we have shown that the function $f$ is bijective for non-parallel vectors. Assume that the input vectors cannot be paralleled. This indicates that for any matrix $Y \in \mathbb{R}^{n \times d}$, there exists a unique $Z \in \mathbb{R}^{n \times d}$ such that $f(Z) = Y$. Next, we are going to show that for a given $Z \in \mathbb{R}^{n \times d}$ and a given $X \in \mathbb{R}^{n \times d}$, there exists a unique set of matrix $(\tilde{W}_Q, \tilde{W}_K, \tilde{W}_V)$ satisfying $\text{rank}(\tilde{W}_Q - W_Q) = s \ll d$, $\text{rank}(\tilde{W}_K - W_K) = s \ll d$ and $\text{rank}(\tilde{W}_V - W_V) = s \ll d$ such that

$$h_A(X; \tilde{W}_Q, \tilde{W}_K, \tilde{W}_V) = Z.$$

We prove it by contradiction. We now assume that there exists a set of matrix

$$(\hat{W}_Q, \hat{W}_K, \hat{W}_V) \neq (\tilde{W}_Q, \tilde{W}_K, \tilde{W}_V).$$

satisfying $\text{rank}(\hat{W}_Q - W_Q) = s \ll d$, $\text{rank}(\hat{W}_K - W_K) = s \ll d$ and $\text{rank}(\hat{W}_V - W_V) = s \ll d$ such that

$$h_A(X; \hat{W}_Q, \hat{W}_K, \hat{W}_V)) = Z.$$

This indicates that

$$\text{softmax}\left( \frac{h_{\text{norm}}(X) R_\theta \tilde{W}_Q \tilde{W}_K^\top R_\theta^\top h_{\text{norm}}^\top(X)}{\sqrt{d_K}} \right) h_{\text{norm}}(X) \tilde{W}_V W_O$$

$$- \text{softmax}\left( \frac{h_{\text{norm}}(X) R_\theta \hat{W}_Q \hat{W}_K^\top R_\theta^\top h_{\text{norm}}^\top(X)}{\sqrt{d_K}} \right) h_{\text{norm}}(X) \hat{W}_V W_O = \mathbf{0}_{n \times d}.$$

Since we assume that the matrix $W_O$ is full-rank, we must have

$$\text{softmax}\left( \frac{h_{\text{norm}}(X) R_\theta \tilde{W}_Q \tilde{W}_K^\top R_\theta^\top h_{\text{norm}}^\top(X)}{\sqrt{d_K}} \right) h_{\text{norm}}(X) \tilde{W}_V$$

$$- \text{softmax}\left( \frac{h_{\text{norm}}(X) R_\theta \hat{W}_Q \hat{W}_K^\top R_\theta^\top h_{\text{norm}}^\top(X)}{\sqrt{d_K}} \right) h_{\text{norm}}(X) \hat{W}_V = \mathbf{0}_{n \times d}. \tag{5}$$

For simplicity of notation, we define

$$\tilde{A}(X) = \text{softmax}\left( \frac{h_{\text{norm}}(X) R_\theta \tilde{W}_Q \tilde{W}_K^\top R_\theta^\top h_{\text{norm}}^\top(X)}{\sqrt{d_K}} \right) h_{\text{norm}}(X)$$

and

$$\hat{A}(X) = \text{softmax}\left( \frac{h_{\text{norm}}(X) R_\theta \hat{W}_Q \hat{W}_K^\top R_\theta^\top h_{\text{norm}}^\top(X)}{\sqrt{d_K}} \right) h_{\text{norm}}(X).$$

We note here that $\tilde{A}(X)$ and $\hat{A}(X)$ are both $n \times n$ matrices, where $n$ denotes the number of input tokens. This further indicates that

$$\tilde{A}(X)\tilde{W}_V - \hat{A}(X)\hat{W}_V = \mathbf{0}_{n \times d}.$$

Consider the case where the input vector $x \in \mathbb{R}^{1 \times d}$ corresponds to a single token, and assume that

$$\text{rank}(x_1, \cdots, x_d) = d.$$

This implies that

$$h_{\text{norm}}(x_1), \cdots, h_{\text{norm}}(x_d) = \omega(x_1)x_1 \odot \gamma, \cdots, \omega(x_d)x_d \odot \gamma,$$

which $\omega(x_i) = \frac{1}{\sqrt{\frac{1}{d}\sum_{j=1}^{d} x_{ij}^2 + \epsilon}}$. By the Lemma 1, this further indicates that

$$\text{rank}(h_{\text{norm}}(x_1), \cdots, h_{\text{norm}}(x_d)) = \text{rank}(x_1, \cdots, x_d) = d,$$

$$\text{softmax}\left(\frac{h_{\text{norm}}(x)R_\theta \hat{W}_Q \hat{W}_K^\top R_\theta^\top h_{\text{norm}}^\top(x)}{\sqrt{d_K}}\right) = 1.$$

Then we have

$$\tilde{A}(x) = \hat{A}(x) = h_{\text{norm}}(x)$$

and

$$(h_{\text{norm}}(x_1), \cdots, h_{\text{norm}}(x_d))(\tilde{W}_V - \hat{W}_V) = \mathbf{0}_d.$$

This shows that

$$\tilde{W}_V = \hat{W}_V$$

Now, redefine our assumption that there exists a set of the matrix such that

$$(\hat{W}_Q, \hat{W}_K) \neq (\tilde{W}_Q, \tilde{W}_K).$$

satisfying $\text{rank}(\hat{W}_Q - W_Q) = s \ll d$ and $\text{rank}(\hat{W}_K - W_K) = s \ll d$ such that

$$h_A(X; \hat{W}_Q, \hat{W}_K, \tilde{W}_V) = Z.$$

We prove it by contradiction. Assume that $X$ is a full rank matrix and $x_i$ cannot lie on the same hyperspace. By using Equation 5, the equation can be transformed into

$$\text{softmax}\left(\frac{h_{\text{norm}}(X)R_\theta \tilde{W}_Q \tilde{W}_K^\top R_\theta^\top h_{\text{norm}}^\top(X)}{\sqrt{d_K}}\right)$$

$$- \text{softmax}\left(\frac{h_{\text{norm}}(X)R_\theta \hat{W}_Q \hat{W}_K^\top R_\theta^\top h_{\text{norm}}^\top(X)}{\sqrt{d_K}}\right) = \mathbf{0}_{n \times d}.$$

By Lemma 3, this indicates that

$$h_{\text{norm}}(X)R_\theta\left(\tilde{W}_Q \tilde{W}_K^\top - \hat{W}_Q \hat{W}_K^\top\right)R_\theta^\top h_{\text{norm}}^\top(X) = \begin{pmatrix} \alpha_1 \\ \alpha_2 \\ \vdots \\ \alpha_n \end{pmatrix}(1, \cdots, 1)_n,$$

For simplicity, define $H(X) = h_{\text{norm}}(X)R_\theta$. The equation can be equivalently transformed into

$$H(X)\boldsymbol{\mu}\boldsymbol{\nu}^\top H^\top(X) = \boldsymbol{\alpha}\mathbf{1}_n^\top.$$

where

$$\boldsymbol{\alpha} = \begin{pmatrix} \alpha_1 \\ \alpha_2 \\ \vdots \\ \alpha_n \end{pmatrix} \quad \text{and} \quad \mathbf{1}_n = \begin{pmatrix} 1 \\ 1 \\ \vdots \\ 1 \end{pmatrix}.$$

Additionally, for $R_\theta$ is full rank matrix and Lemma 1,

$$\text{rank}(H(X)) = \text{rank}(X) = d.$$

This further implies two cases:

**Case 1: $\boldsymbol{\alpha} = \mathbf{0}_n$**

$$H(X)\boldsymbol{\mu}\boldsymbol{\nu}^\top H^\top(X) = \mathbf{0}_{n\times n}.$$

Since $H(X)$ is a full-rank matrix, we must have that

$$\tilde{W}_Q\tilde{W}_K^\top - \hat{W}_Q\hat{W}_K^\top = \mathbf{0}_{d\times d},$$

which means that

$$\tilde{W}_Q\tilde{W}_K^\top = \hat{W}_Q\hat{W}_K^\top$$

**Case 2:$\boldsymbol{\alpha} \neq \mathbf{0}_n$**

Suppose that $X$ is full rank, we must have $\text{rank}(\tilde{W}_Q\tilde{W}_K^\top - \hat{W}_Q\hat{W}_K^\top) = 1$. This indicates that

$$\tilde{W}_Q\tilde{W}_K^\top - \hat{W}_Q\hat{W}_K^\top = \boldsymbol{\mu}\boldsymbol{\nu}^\top,$$

which

$$\boldsymbol{\mu} = \begin{pmatrix} \mu_1 \\ \mu_2 \\ \vdots \\ \mu_n \end{pmatrix} \text{ and } \boldsymbol{\nu} = \begin{pmatrix} \nu_1 \\ \nu_2 \\ \vdots \\ \nu_n \end{pmatrix}.$$

We can get

$$H(X)\boldsymbol{\mu}\boldsymbol{\nu}^\top (H(x_1)^\top, \cdots, H(x_n)^\top) = \boldsymbol{\alpha}\mathbf{1}_n^\top$$

and

$$H(X)\boldsymbol{\mu} = \frac{1}{\beta}\boldsymbol{\alpha} \text{ and } \boldsymbol{\nu}^\top (H(x_1)^\top, \cdots, H(x_n)^\top) = \beta\mathbf{1}_n^\top,$$

which indicates that

$$h_{\text{norm}}(x_1)R_\theta\boldsymbol{\nu} = h_{\text{norm}}(x_2)R_\theta\boldsymbol{\nu} = \cdots = h_{\text{norm}}(x_n)R_\theta\boldsymbol{\nu} = \beta.$$

This indicates that

$$\begin{cases} x_1 = \frac{\beta\sqrt{\frac{1}{d}\sum_{j=1}^d x_{1j}^2+\epsilon}}{\gamma}(R_\theta\boldsymbol{\nu})^{-1} \\ x_2 = \frac{\beta\sqrt{\frac{1}{d}\sum_{j=1}^d x_{2j}^2+\epsilon}}{\gamma}(R_\theta\boldsymbol{\nu})^{-1} \\ \vdots \\ x_n = \frac{\beta\sqrt{\frac{1}{d}\sum_{j=1}^d x_{nj}^2+\epsilon}}{\gamma}(R_\theta\boldsymbol{\nu})^{-1} \end{cases}$$

Suppose that $(x_1, \cdots, x_d)$spans an independent linear space,we must have that

$$x_{d+i} = c_{(d+i)1}x_1 + \cdots + c_{(d+i)d}x_d \quad i \in (1, \cdots, n-d),$$

which is contradictory. Thus, we conclude that

$$\boldsymbol{\alpha} = \mathbf{0}_n,$$

which indicates that

$$\tilde{W}_Q\tilde{W}_K^\top = \hat{W}_Q\hat{W}_K^\top$$

$\square$

# B EXPERIMENTS

## B.1 MODEL DETAILS

We selected several base models from the LLaMA and Mistral families. The models included different sizes: 7B, 13B, and 70B for LLaMA 2, 8B and 70B for LLaMA 3, and Mistral 7B. Each of these models was used as a base for LoRA fine-tuning. The rank of LoRA fine-tuning ranged from 8 to 256, with common ranks being 8, 16, 128, and 256. The fine-tuning domains encompassed finance, legal, and medical fields.

## B.2 DATASETS

The dataset utilized in this experiment consists of 5,579 words, all selected from the vocabularies of various models. Each word can be recognized as a complete token by the model's tokenizer and is subsequently transformed into a vector representation with dimensionality *d_model* through the model's embedding layer. The selection of these words ensures their frequent appearance in the respective vocabularies, and each word's embedding can be used for further model training and evaluation.

The table below outlines the average, minimum, and maximum word lengths for each model:

| Model | Average | Minimum | Maximum |
|---|---|---|---|
| **LLaMA2** | 5.8 | 1 | 15 |
| **LLaMA3** | 5.5 | 1 | 36 |
| **Mistral** | 6.1 | 1 | 14 |

Table 2: Word lengths for different models

## B.3 EXPERIMENT DETAILS

We reconstruct MLP inputs using gradient descent with the Adam optimizer (initial learning rate of 1.5e-3), following a process of 700 iterations in StepLR scheduler with a step size of 1 with a gamma of 0.9999 for gradual learning rate decay, balancing rapid convergence and fine-tuning; furthermore, to address incorrect values in reverse-engineered MLP inputs that may inflate rank estimates during SVD of LoRA fine-tuned outputs, we conduct 50 iterations of stochastic sampling (each generating 520 outputs) and use the minimum rank from these iterations to improve the robustness of the final rank estimation by minimizing the effect of outliers.

