# OpenReview forum: "Toward Trustworthy: A Method for Detecting Fine-Tuning Origins in LLMs"
_ICLR.cc/2025/Conference — Submitted to ICLR 2025_

### Official Review · Reviewer_YHDG · 2024-10-18

**Soundness:** 2
**Presentation:** 2
**Contribution:** 2
**Rating:** 5
**Confidence:** 3

**Summary:**

This paper introduces a new method to verify whether large language models (LLMs) have been fine-tuned from a specified base model, addressing limitations in existing verification techniques. The approach can detect obfuscation tactics, such as permutations and scaling, that obscure a model’s origin. Additionally, the framework extracts the LoRA rank used during fine-tuning, offering a more robust verification system. The method is empirically validated on 29 diverse models, showing its effectiveness in challenging real-world scenarios.

**Strengths:**

- The problem is very important.
- Handling obfuscation, making it robust against manipulation.
- A formal framework focused on identifying fine-tuning origins.
- Extensive validation across multiple models.

**Weaknesses:**

- Missing related work and discussion: The procedure (line 279) in Algorithm 1 "Random Rank Extraction" seems very similar to the counterpart in Algorithm 1 "Hidden-Dimension Extraction Attack" in the paper [1]. The "Rank Extraction Method" is also similar to this paper, but the authors never reference it in the submission. Please provide more discussion about it.
- Strong assumption in the proof: The assumption in line 742 seems strong. Is there any support for this full rank assumption? Such issue is the same for the assumption in line 785. Unfortunately, some of these assumtions are not even shown in the main paper. They only appear in the appendix. I believe that some important assumptions should be clearly explained in the main paper. The assumption of linear independency in line 217 is also too strong.
- Unreasonable insight: I question the rationality of the statement "if outputs are nearly identical, their corresponding intermediates are likely similar" in line 269. It's intuitive but not rigid enough.

If the author's answers address my concerns, I will consider raising the score.

[1] Carlini N, Paleka D, Dvijotham K D, et al. Stealing part of a production language model. ICML 2024.

**Questions:**

Typos and some minor issues:
- The caption of figure 2 should be modified to prevent the overlap.
- $W$ in line 168 should be revised, is it $W_c$?
- "scalara" in line 200.
- You should leave a blank space between $R_\theta$ and "is" in line 186. There is also a similar issue in line 194 and 200.
- There should be a reference in line 223 for Natural Language Toolkit (NLTK).
- None of the formula in this paper is numbered.
- What is the symbol between $x_i$ and $y_i$ in line 575? I suggest the authors to incorporate a formal definition of it since this symbol is not as routine to the reader as addition (+), subtraction (-), multiplication (×), and division (÷).
- The full name of "MLP" should be introduced in this paper.

Others:
- I believe that there should be some reference in Section 3.3 to help readers better understand the problem. We cannot guarantee that all readers are familiar with this field. In particular, the readers may be confused about the sentence "The challenge posed by this scenario is encapsulated by the discrepancy in ranks of the parameter differences".
- There should be some reference for PEFT in line 86.

---

### Official Review · Reviewer_DVBE · 2024-10-28

**Soundness:** 2
**Presentation:** 1
**Contribution:** 2
**Rating:** 3
**Confidence:** 4

**Summary:**

This paper aims to propose a method for rigorously determining whether a model has been fine-tuned from a specified base model.

**Strengths:**

This is an important task, and the experiments appear to be thorough.

**Weaknesses:**

The presentation of this paper is poor, which affects my overall understanding. For example, I don't understand the purpose of each subsection of the methodology. The writing is difficult to follow, making it challenging to grasp the key points. The methodology section is particularly confusing, as the steps are not well-explained, further hindering my comprehension of the paper. The experimental analysis should be more thorough. I would like to obtain a more detailed quantitative analysis.

The use of the term "trustworthy" in the title is unclear, leaving its meaning ambiguous. The title is also grammatically not good.

I also question the claim that "Crucially, the method remains valid regardless of the permutations used, enabling accurate determination of the base model for any derivative." If the parameters are significantly altered, it becomes theoretically impossible to determine whether the model was trained from scratch or fine-tuned. This differs significantly from the problem definition, which is based on LoRA. Could the authors clarify more about this?

**Questions:**

NA

---

### Official Review · Reviewer_7uHp · 2024-11-03

**Soundness:** 3
**Presentation:** 3
**Contribution:** 3
**Rating:** 6
**Confidence:** 3

**Summary:**

This paper introduces a novel method for detecting fine-tuning origins in large language models (LLMs), addressing the challenge of transparency and trust when obfuscation techniques are used to hide model lineage. The method, which is the first of its kind, can extract the Low-Rank Adaptation (LoRA) rank used during fine-tuning, providing a robust verification framework. It has been empirically validated on 29 diverse open-source models, demonstrating its effectiveness in real-world scenarios. The study contributes to enhancing the trustworthiness and accountability of AI model deployments by accurately documenting model origins and modifications.

**Strengths:**

Robustness: the ability of the method to recognize models that are fine-tuned even in the presence of obfuscation techniques (e.g., parameter permutations and scaling transformations) demonstrates its robustness against common means of obfuscation.

Accuracy: By extracting the LoRA rank, the method is able to accurately identify the differences between the fine-tuned model and the base model, providing a detailed quantitative measure of the origin of the model.

**Weaknesses:**

Scope limitation: The current method is mainly applicable to the case where the MLP layer is not modified during the fine-tuning process. If the parameters of the MLP layer are tuned or the architecture is changed, the effectiveness of the method may be reduced.

Challenges of small-amplitude output models: for models with small output amplitudes, the efficiency of reverse-engineering intermediate states may suffer due to weak gradient signals, which limits the applicability of the method.

Computational complexity: the use of techniques such as SVD and gradient descent may entail high computational costs, especially when dealing with large models.

Lack of discussion with backdoor-based detection approach like [1]

[1] Double-I Watermark: Protecting Model Copyright for LLM Fine-tuning

**Questions:**

The paper mentions that the methodology is mainly applicable to the case where the MLP layer has not been modified. How will future research be extended to accommodate cases where the MLP layer parameters are adjusted, or the architecture is changed?

How well does the paper's approach generalize to different types of models and fine-tuning strategies? Are there plans for more extensive experiments to validate this?

In the iterative optimization strategy, the update formula $y_{m+1} =y_{m} -\alpha \nabla \| f(y_{m} )-z_{c} \| ^{2}$ depends on the gradient$ \nabla \| f(y_{m} )-z_{c} \| ^{2}$. If $f(y_{m} )$ is not smooth in the neighborhood around $y_{m}$, will this affect the stability of the gradient and the convergence of the reconstruction process? In the presence of multiple local minima, how to ensure that the gradient descent method can find the global minimum and not get stuck in the local minimum, especially when $z_{c}$ is affected by obfuscation techniques?

---

### Official Review · Reviewer_4SLG · 2024-11-11

**Soundness:** 3
**Presentation:** 2
**Contribution:** 2
**Rating:** 3
**Confidence:** 4

**Summary:**

The paper proposes a technique to identify if a target model is a finetuned version of a base model, when the target model can be obfuscated. Gray box access to intermediate model weights is assumed.

**Strengths:**

1. I believe model attribution issues will only rise with the craze related to ML. Hence this paper is timely.
2. For open source models, model obfuscation is perhaps an important way of hiding the base model information. Hence the focus on obfuscation is good.
3. The paper considers many models in its experiments section including llama2, 3 and Mistral models.

**Weaknesses:**

1. The paper really needs to be rewritten. Here are some comments:
    a) improve the caption of Fig.1 -- readers are supposed to understand this figure without reading the paper.
    b) Fig 2 caption and text are mixed.
    c) how are sections 4.1, 4.2 and 4.3 connected? -- it would help to give a para before 4.1 to explain the same.
    d) Alg 1 should be self-sufficient -- mention what each notation means in the algorithm itself.
    e) Where is theorem 4? where does the proof end? denote with end of proof symbol.
    f) The captions of all figures should be better.
    g) The explanation of results is very weak.
    h) there should be more focus on sections 4 and 5.2.

2. The paper is really specific to fine tuning. Not sure if it works for distillation and compression.
3. L49-51 where the authors claim to be the "first formal framework" seems a bit of an over claim. Techniques such as using fingerprinting have been shown to last for fine tuning (that too with black box access).
4. I am not sure if the technique really works in non-trivial cases. a) the paper excludes obfuscation in the MLP layers -- why is this a reasonable assumption? won't it be easy for an adversary to manipulate MLP layers? b) When the LoRA config includes W_o , the technique completely fails , by estimating ranks very off from the given rank.

**Questions:**

1) Why is rank(Wc - Wb) >> s?

---

### Meta-Review · Area_Chair_mPbE · 2024-12-19

**Metareview:**

While the reviewers appreciated the paper’s  motivation and initial experiments, their main concerns were with (a) the scope of the method, (b) the computational complexity, (c) missing related work discussion, (d) the strength of required assumptions, and (e) overall clarity. There was no author response. For these reasons I vote to reject. The reviewers have given detailed feedback and I recommend the authors follow / respond to their comments closely before submitting to a future ML venue. If the authors are able to fix these things it will make a much stronger submission.

**Additional Comments On Reviewer Discussion:**

There was no reviewer discussion and no author rebuttal. All reviewers except one voted to reject and the concerns of this reviewer were not addressed.

---

### Decision · Program_Chairs · 2025-01-22

Reject